# Factors associated with contrast-associated acute kidney injury in an emergency department: A cohort study in Lebanon

**Moustafa Al Hariri[1], Sally Al Hassan[2], Malak Khalifeh[2], Hani Tamim[3,4], Imad El Majzoub[5], Tharwat El Zahran[2]\***

1 Tamayuz Simulation Center, QU Health Sector, Qatar University, Doha, Qatar, 2 Department of Emergency Medicine, American University of Beirut Medical Center, Beirut, Lebanon, 3 Department of Internal Medicine, American University of Beirut Medical Center, Beirut, Lebanon, 4 College of Medicine, Alfaisal University, Riyadh, Saudi Arabia, 5 Department of Emergency Medicine, Sheikh Shakhbout Medical City, Abu Dhabi, United Arab Emirates

\* Te15@aub.edu.lb

## Abstract

### Background

Contrast-associated acute kidney injury (CA-AKI) is a common problem in hospitals, particularly in low-middle-income countries (LMIC), due to limited resources and a high prevalence of comorbidities. Kidney function evaluation using serum creatinine levels before contrast administration leads to increased length of stay and delayed patient care. This study aimed to identify factors associated with CA-AKI in emergency department (ED) patients in an LMIC. Identifying these factors is essential for enhancing patient care and guiding clinical practice by allowing for the early detection and management of patients at risk.

### Methods

This study is a retrospective cohort study conducted at the largest tertiary care center's ED in Lebanon between November 2018 and December 2019. The study included ED patients who underwent computed-tomography (CT) with contrast. Bivariate and logistic regression analyses were performed to compare the characteristics of patients who developed AKI with those who did not by using SPSS package. The Institutional Review Board (IRB) at the American University of Beirut (AUB) approved this study under protocol ID BIO-2020-0276, which was performed per the Declaration of Helsinki. The IRB waived the need to consent patients since many of them were not followed up at the time of the study.

### Results

The study included 1832 patients, of whom 10.4% (n = 190) developed CA-AKI. Patients aged over 65 had a 1.6-fold higher risk of CA-AKI (aOR = 1.55, 95%CI:1.09-2.2). High blood pressure (≥140 mmHg), high respiratory rate (≥ 22), and chronic kidney disease were significantly associated with CA-AKI. The use of loop diuretics (aOR = 2.21, 95%CI:1.49-3.28),

**Data availability statement:** The datasets generated during and/or analysed during the current study are publically available through this link: https://qspace.qu.edu.qa/handle/10576/57874

**Funding:** The author(s) received no specific funding for this work.

**Competing interests:** The authors have declared that no competing interests exist.

**Abbreviations:** AKI: Acute Kidney Injury; aOR: Adjusted Odds Ratio; AUB: American University of Beirut; AUBMC: American University of Beirut Medical Center; ACEI: Angiotensin-Converting Enzyme Inhibitors; CA-AKI: Contrast-associated Acute Kidney Injury; CT: Computed-Tomography; CIN: Contrast-Induced Nephrotoxicity; DEM: Department Of Emergency Medicine; ED: Emergency Department; eGFR: Estimated Glomerular Filtration Rate; IRB: Institutional Review Board; IQR: Interquartile Range; KDIGO: Kidney Disease Improving Global Outcomes; LMIC: Low-Middle-Income Countries; NSAID: Non-Steroidal Anti-Inflammatory Drug; RR: Respiratory Rate; STROBE: Strengthening the Reporting of Observational Studies in Epidemiology; SBP: Systolic Blood Pressure.

beta-lactams (aOR = 4.11, 95%CI:2.63-6.42), and allopurinol (aOR = 2.74, 95%CI:1.43-5.25) were significantly associated with CA-AKI.

## Conclusions

Identifying factors associated with CA-AKI in an emergency setting, such as age, comorbidities, and home medications, can help identify patients at low risk of developing CA-AKI.

## Background

Contrast-associated acute kidney injury (CA-AKI) is a critical concern in healthcare due to its significant impact on patient outcomes and healthcare costs. CA-AKI is the third most prevalent cause of hospital-acquired Acute Kidney Injury (AKI), affecting up to 30% of patients who receive iodinated contrast media [1]. This condition can lead to prolonged hospital stays, increased morbidity and mortality, and higher healthcare costs. The pathogenesis of CA-AKI involves contrast-induced vasoconstriction, which reduces renal medullary blood flow, exacerbated by comorbidities such as diabetes mellitus, congestive heart failure, and chronic kidney disease [2,3]. Risk factors include advanced age, high blood pressure, high respiratory rate, and the use of nephrotoxic medications like loop diuretics and beta-lactams [4,5]. Preventive strategies, such as pre-procedural hydration and minimizing contrast dose, are essential to mitigate the risk of CA-AKI [6]. Given the administration of over 30 million doses of iodinated contrast medium annually, the burden of CA-AKI on the healthcare system is substantial [1]. Effective management protocols and early identification of at-risk patients are crucial to improving outcomes and reducing costs associated with CA-AKI.

In 2020, the American College of Radiology and the National Kidney Foundation released joint guidelines distinguishing between CA-AKI and contrast-induced acute kidney injury (CI-AKI). CA-AKI refers to AKI that occurs in temporal association with contrast media administration but does not imply causation, whereas CI-AKI implies a direct causal relationship between contrast media and AKI [2].

The pathogenesis behind CA-AKI is contrast-induced vasoconstriction, which decreases renal medullary blood flow. This vasoconstriction is triggered by the release of endothelin and adenosine, leading to reduced oxygen delivery to the renal tissue [3]. Comorbidities such as diabetes mellitus, congestive heart failure, acute hypotension (requiring pressors or intra-aortic balloon pump), ST-elevation myocardial infarction, and volume depletion in patients at risk for CA-AKI worsen this condition by further impairing renal perfusion and increasing oxidative stress [5]. For instance, diabetes mellitus can lead to endothelial dysfunction, while congestive heart failure can cause reduced cardiac output, both of which aggravate renal hypoxia. Acute hypotension, often requiring pressors or intra-aortic balloon pump, can lead to further renal ischemia. Patients with chronic kidney disease are particularly vulnerable due to their reduced nephron number and weakened compensatory mechanisms [4]. Additional risk factors include procedural difficulties, such as the volume and type of contrast used [1]. Risk factor scoring systems of both baseline comorbidities and procedural factors have been used as predictors for CA-AKI, the need for renal replacement therapy, and long-term mortality [7].

One effective strategy to prevent CA-AKI is volume expansion through intravenous fluid infusion, which helps increase urine flow and dilute the contrast medium. This method is effective because it enhances renal perfusion and reduces the concentration of contrast agents in the renal tubules, thereby minimizing nephrotoxic effects [8]. In hospitalized patients,

this can be achieved by initating intravenous infusion 6-12 hours before contrast adminis-tration and counting for at least 4 hours afterwards [6]. However, in the emergency setting, this approach is challenging due to the need for immediate imaging and prompt diagnosis of life-threatening conditions [1]. Alternative strategies include using low-osmolality or iso-osmolality contrast agents, which are less nephrotoxic, and administering medications such as N-acetylcysteine to reduce oxidative stress [9,10].

Measuring serum creatinine is the standard method to assess kidney function before contrast administration. Nevertheless, measuring serum creatinine for every Emergency Department (ED) patient adds a financial burden on the patient, increases the wait time for prompt diagnosis, increases the length of ED stay, and delays care [8]. To minimize these issues while ensuring patient safety, alternative strategies include using risk assessment ques-tionnaires to identify patients at low risk of CA-AKI, thereby reducing unnecessary creatinine testing [8]. Additionally, point-of-care testing for creatinine levels can provide rapid results, facilitating quicker decision-making. Implementing electronic health record (EHR) alerts for patients with known risk factors can also help streamline the process and ensure timely interventions[11,12].

It is essential to differentiate between CA-AKI and CI-AKI. CA-AKI is a correlative diagnosis indicating that AKI occurred after contrast administration, while CI-AKI implies a causative relation between contrast media and AKI [2]. This distinction is crucial because our study design does not allow for causal inference.

## Methods

### Aim

This study aims to assess the factors associated with the development of CA-AKI in patients who received computed tomography (CT) scans with contrast during their presentation to an academic Emergency Department in a tertiary care center.

### Study design and setting

This study was an IRB-approved, retrospective, observational cohort study conducted between November 2018 and December 2019. This study was conducted at the Department of Emergency Medicine (DEM) of the American University of Beirut Medical Center (AUBMC), the largest academic and tertiary care center in Lebanon, a low-middle-income country. The Institutional Review Board (IRB) at the American University of Beirut (AUB) approved this study under protocol ID BIO-2020-0276 in July 2020 and the study was performed in accor-dance with the Declaration of Helsinki. The medical record team at AUBMC provided the authors with the patient list and data in August 2021. Although the exported data contained the identifiers of participants (medical record number, and date and time of ED arrival) we did not have to access the medical records of the patients since the exported data by the med-ical record team contained all needed study variables. No participant identifiers are included in this manuscript. This study conforms to the Strengthening the Reporting of Observational Studies in Epidemiology (STROBE) guidelines.

### Population

Patients presenting to the DEM who received Computed Tomography (CT) with contrast (CI-CT) as a clinical workup were identified by the Electronic Health Record (EPIC) team in August 2021. An average of 2ml/kg of contrast solution was administered for the study partic-ipants. We excluded the records of 328 patients that lacked baseline creatinine level and lacked creatinine level assessment within 48 hours post CI-CT. Due to the retrospective nature of the

study, patients were not involved in its design.. However, we plan to share the study's findings with healthcare professionals through different routes.

## Outcome measures

Patients' details were prepared as reports by the EPIC team. Data management and merging were performed to collate all participants' variables longitudinally. Participants' variables of interest included demographics, past medical and surgical history, home medication, vitals, ED management (medications and IV administration), home medications, and laboratory workup. We included the records of the patients up to two days before the contrast administration and for a maximum of 7 days following contrast administration to collect the data on the IV-fluid administration and creatinine and/or eGFR laboratory values to be able to assess AKI. Cutoffs for Systolic blood pressure (≥140 mmHg) [13], Heart rate (≥100 bps) [14], and respiratory rate (≥22 bps) [15] were implemented in the analysis according to the values defined in our institution and previous reports in the literature.

## AKI definitions

This study used the Kidney Disease Improving Global Outcomes (KDIGO) definition for AKI, which defines AKI as an increase in creatinine level of 0.3 mg/dl (26 mmol/l) within 48 hours or a 50% increase over the baseline level within 7 days [16]. The records of the identified participants who did not have at least one creatinine measurement within 48 hours of CI-CT were excluded from the study.

## Sample size

A total of 1832 participants were included who fulfilled the inclusion criteria..

## Data analysis

Data analysis was performed using SPSS (IBM SPSS 25.0). Categorical variables were presented using frequencies and percentages, while continuous variables were presented using mean and standard deviation (mean ± SD). To observe the associations between the different categorical variables and our outcome, Pearson's Chi-square and Fisher's exact tests, where applicable, were used for categorical variables and Student's T-test for continuous ones. To account for multiple comparisons, q values (FDR) were calculated and reported.

Logistic regression was then performed to adjust confounding variables and test the association between contrast-associated AKI and other variables. Variables that showed a significant confidence interval of the unadjusted OR and clinically relevant were selected to build the logistic regression model. Significance was interpreted at α ≤ 0.05.

# Results

## Baseline Characteristics

In this study, a total of 1832 patients were included, among which 10.4% developed CA-AKI (n = 190). The median age of the participants was 60 years old (IQR: 42.39-74), and 51.7% were females (n = 948). Only 43% reported being smokers (n = 787), and 25.8% reported being alcohol consumers (n = 189). Cardiovascular diseases were the most prevalent medical history (n = 538, 29.4%) followed by cancer (n = 651, 35.5%), hypertension (n = 333, 18.2%), diabetes (n = 81, 4.4%) and liver diseases (n = 132, 7.2%) (Table 1).

**Table 1. Association of baseline characteristics and AKI.**

|  | options | Total N = 1832 | No AKI N = 1642 (%) | AKI N = 190 (%) | p-value | q-Value | OR | 95%CI |
|---|---|---|---|---|---|---|---|---|
| **Age** | **< 60 years** | 915 (49.9%) | 854 (52%) | 61 (32.1%) | < 0.001 | <0.001 | Ref | |
|  | **≥ 60 years** | 917 (50.1%) | 788 (48%) | 129 (67.9%) | | | 2.29 | 1.67-3.16 |
| **Gender** | **Female** | 948 (51.7%) | 858 (52.3%) | 90 (47.4%) | 0.202 | 0.358 | Ref | |
|  | **Male** | 884 (48.3%) | 784 (47.7%) | 100 (52.6%) | | | 1.22 | 0.9-1.64 |
| **Alcohol** | | 189 (25.8%) | 172 (25.8%) | 17 (25.8%) | 0.512 | 0.713 | 1.00 | 0.56-1.78 |
| **Smoking** | | 787 (43%) | 715 (43.5%) | 72 (37.9%) | 0.136 | 0.280 | 0.79 | 0.58-1.08 |
| **Comorbidities:** | **Cardiovascular diseases** | 538 (29.4%) | 455 (27.7%) | 83 (43.7%) | < 0.001 | <0.001 | 2.02 | 1.49-2.75 |
|  | **Hypertension** | 333 (18.2%) | 288 (17.5%) | 45 (23.7%) | 0.038 | 0.086 | 1.46 | 1.02-2.09 |
|  | **Diabetes** | 181 (9.9%) | 154 (9.4%) | 27 (14.2%) | 0.035 | 0.084 | 1.60 | 1.03-2.48 |
|  | **Cancer** | 651 (35.5%) | 575 (35%) | 76 (40%) | 0.174 | 0.340 | 1.24 | 0.91-1.68 |
|  | **Rheumatological disease** | 38 (2.1%) | 37 (2.3%) | 1 (0.5%) | 0.173 | 1.000 | 0.23 | 0.03-1.68 |
|  | **Chronic Kidney Disease** | 81 (4.4%) | 53 (3.2%) | 28 (14.7%) | < 0.001 | <0.001 | 5.18 | 3.19-8.42 |
|  | **Organ Transplant** | 67 (3.7%) | 58 (3.5%) | 9 (4.7%) | 0.402 | 0.654 | 1.36 | 0.66-2.79 |
|  | **Liver disease** | 132 (7.2%) | 110 (6.7%) | 22 (11.6%) | 0.014 | 0.041 | 1.82 | 1.12-2.96 |

Data are presented as numbers with percentages.

The p-value for the difference between AKI vs. no AKI is calculated by chi-square, Fisher´s exact test, or T-test where appropriate.

The q-value is the False Discovery Rate (FDR) significance accounting for multiple comparisons.

The unadjusted Odd Ratio (OR) for each variable was calculated without accounting for any other factor. 95% CI is the confidence interval of the OR for each variable.

Additionally, most participants had normal baseline eGFR (≥65 ml/min) (n = 1540, 87.6%). A minority had a high qSOFA score of either 1 (20.1%) or 2 (2.3%). Table 2 shows different concomitant nephrotoxic medications taken by the patients. About 22.9% were on aspirin (n = 419),12.2% were on loop diuretics (n = 224), 12.2% on vancomycin (n = 224) followed by thiazide diuretic (n = 140, 7.6%), beta-lactam antibiotics (n = 134, 7.3%), allopurinol (n = 63, 3.4%), and NSAIDs (n = 47, 2.6%) (Table 2).

## Characteristics of patients with contrast-associated AKI

Patients with AKI were older than 65 (58.9% vs. 37.4%, p < 0.001). However, there was no significant difference in gender, smoking status, and alcohol consumption status with CA-AKI. As for the types of comorbidities associated with CA-AKI, cardiovascular diseases (43.7% vs. 27.7%, p < 0.001) showed a significant association. Other comorbidities showed significant association with CA-AKI, including hypertension (23.7% vs. 17.5%, p = 0.038), diabetes (14.2% vs. 9.4%, p = 0.035), chronic kidney disease (14.7% vs. 3.2%, p < 0.001), and liver diseases (11.6% vs. 6.7%, p = 0.014) (Table 1).

As for vital signs, systolic blood pressure and respiratory rate were significantly associated with CA-AKI. Patients with high blood pressure (≥140 mmHg) and high respiratory rate (≥ 22) were more associated with CA-AKI.

As summarized in Table 2, a significantly higher portion of the CA-AKI patients reported to be on loop diuretics (32.1% vs. 9.9%, p < 0.001), aminoglycosides (6.8% vs. 3.2%, p = 0.011), beta-lactams (26.3% vs. 5.1%, p < 0.001), allopurinol (8.4% vs. 2.9%, p < 0.001), lithium (30.5% vs. 22%, p = 0.008), aspirin (24.7% vs. 10.8%, p < 0.001), and colistin (7.4% vs. 1.3%, p < 0.001). Patients with CA-AKI had significantly higher qSOFA score either 1 (35.8% vs. 18.3%, p < 0.001) or 2 (3.7% vs. 2.3%, p < 0.001). The percentages of patients with CA-AKI were higher in patients with lower baseline eGFR, as shown in Fig 1 (p < 0.001).

**Table 2. Association of clinical characteristics and AKI.**

| | options | Total N = 1832 | No AKI N = 1642 (%) | AKI N = 190 (%) | p-value | q-value | OR | 95%CI |
|---|---|---|---|---|---|---|---|---|
| qSOFA | 0 | 1419 (77.6%) | 1305 (79.6%) | 114 (60%) | < 0.001 | <0.001 | Ref | |
| | 1 | 368 (20.1%) | 300 (18.3%) | 68 (35.8%) | | | 2.60 | 1.87-3.59 |
| | 2 | 42 (2.3%) | 34 (2.1%) | 8 (4.2%) | | | 2.69 | 1.22-5.96 |
| Systolic Blood Pressure (SBP) | < 140 | 1218 (66.6%) | 1107 (67.5%) | 111 (58.4%) | 0.012 | 0.038 | Ref | |
| | ≥ 140 | 611 (33.4%) | 532 (32.5%) | 79 (41.6%) | | | 1.48 | 1.09-2.01 |
| Heart Rate (HR) | < 100 | 1111 (60.7%) | 994 (60.6%) | 117 (61.6%) | 0.803 | 1.000 | Ref | |
| | ≥ 100 | 718 (39.3%) | 645 (39.4%) | 73 (38.4%) | | | 0.96 | 0.71-1.31 |
| Respiratory Rate (RR) | < 22 | 1531 (83.7%) | 1404 (85.7%) | 127 (66.8%) | < 0.001 | <0.001 | Ref | |
| | ≥ 22 | 298 (16.3%) | 235 (14.3%) | 63 (33.2%) | | | 2.96 | 2.13-4.13 |
| Received IV Hydration prior to CT | | 1800 (98.3%) | 1613 (98.2%) | 187 (98.4%) | 1.000 | 1.000 | 0.85 | 0.34-3.71 |
| Performed Angio CT | | 436 (23.8%) | 378 (23.0%) | 58 (30.5%) | 0.022 | 0.056 | 1.47 | 1.06-2.04 |
| Concomitant Nephrotoxic Medications: | **Antiviral** | 41 (2.2%) | 34 (2.1%) | 7 (3.7%) | 0.187 | 0.348 | 1.81 | 0.79-4.14 |
| | **Antiretroviral** | 1 (0.1%) | 1 (0.1%) | 0 (0%) | 1.000 | 1.000 | | |
| | **Loop Diuretics** | 224 (12.2%) | 163 (9.9%) | 61 (32.1%) | < 0.001 | <0.001 | 4.29 | 3.04-6.06 |
| | **Thiazide Diuretics** | 140 (7.6%) | 123 (7.5%) | 17 (8.9%) | 0.474 | 0.712 | 1.21 | 0.71-2.06 |
| | **ACEI** | 55 (3%) | 51 (3.1%) | 4 (2.1%) | 0.444 | 0.693 | 0.67 | 0.24-1.88 |
| | **Chemotherapy\*** | 12 (0.7%) | 11 (0.7%) | 1 (0.5%) | 1.000 | 1.000 | 0.79 | 0.10-6.11 |
| | **Immunotherapy\*\*** | 7 (0.4%) | 7 (0.4%) | 0 (0%) | 1.000 | 1.000 | | |
| | **NSAID** | 47 (2.6%) | 43 (2.6%) | 4 (2.1%) | 1.000 | 1.000 | 0.8 | 0.28-2.25 |
| | **Aminoglycoside** | 66 (3.6%) | 53 (3.2%) | 13 (6.8%) | 0.016 | 0.044 | 2.20 | 1.18-4.12 |
| | **Beta Lactam** | 134 (7.3%) | 84 (5.1%) | 50 (26.3%) | < 0.001 | <0.001 | 6.62 | 4.48-9.79 |
| | **Sulfonamides** | 18 (1.0%) | 14 (0.9%) | 4 (2.1%) | 0.108 | 0.234 | 1.07 | 0.58-2.00 |
| | **Antiepileptic** | 26 (1.4%) | 25 (1.5%) | 1 (0.5%) | 0.511 | 0.731 | 0.34 | 0.05-2.54 |
| | **Allopurinol** | 63 (3.4%) | 47 (2.9%) | 16 (8.4%) | < 0.001 | <0.001 | 3.12 | 1.73-5.62 |
| | **Dapsone** | 2 (0.1%) | 2 (0.1%) | 0 (0%) | 1.000 | 1.000 | | |
| | **Amitryptilline** | 16 (0.9%) | 14 (0.9%) | 2 (1.1%) | 0.678 | 0.912 | 1.24 | 0.28-5.49 |
| | **Lanzoprazole** | 9 (0.5%) | 8 (0.5%) | 1 (0.5%) | 1.000 | 1.000 | 1.08 | 0.13-8.69 |
| | **Rifampicin** | 3 (0.2%) | 2 (0.1%) | 1 (0.5%) | 0.28 | 0.475 | 4.34 | 0.39-48.07 |
| | **Tetracycline** | 2 (0.1%) | 2 (0.1%) | 0 (0%) | 1.000 | 1.000 | | |
| | **Lithium** | 6 (0.3%) | 6 (0.4%) | 0 (0%) | 1.000 | 1.000 | | |
| | **Aspirin** | 419 (22.9%) | 361 (22%) | 58 (30.5%) | 0.008 | 0.028 | 1.56 | 1.12-2.17 |
| | **Vancomycin** | 224 (12.2%) | 177 (10.8%) | 47 (24.7%) | < 0.001 | < 0.001 | 2.72 | 1.89-3.92 |
| | **Colistin** | 36 (2%) | 22 (1.3%) | 14 (7.4%) | < 0.001 | < 0.001 | 5.86 | 2.94-11.65 |

Data are presented as numbers with percentages for categorical variables and Median and [Interquartile ranges] for continuous variables.

\* Antiviral included Acyclovir, Cidofovir, Foscarnet, and Valganciclovir. \*\*Chemotherapy includes Cisplatin, Carboplatin, Cyclophosphamide, Ifosfamide, and Methotrexate. \*\*\* Immunotherapies include Sirolimus, Tacrolimus, and Cyclosporine.

Abbreviations: ACEI: Angiotensin-converting enzyme Inhibitor and NSAID: Non-steroidal anti-inflammatory drugs.

The p-value for the difference between AKI vs. no AKI is calculated by chi-square, Fisher´s exact test, or T-test where appropriate.

The q-value is the False Discovery Rate (FDR) significance accounting for multiple comparisons.

The unadjusted Odd Ratio (OR) for each variable was calculated without accounting for any other factor. 95% CI is the confidence interval of the OR for each variable.

## Factors associated with contrast-associated AKI

After adjusting for confounding variables using logistic regression, patients greater than 65 years old were about 1.6 times more associated with CA-AKI (aOR = 1.55, 95%CI: 1.09-2.2). Patients with high blood pressure (≥140 mmHg) and high respiratory rate (≥ 22) were more

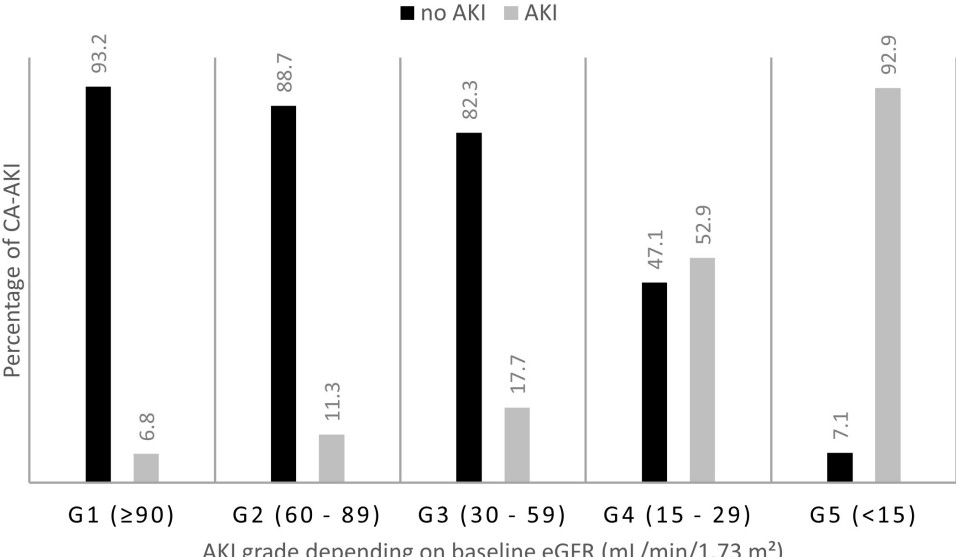

**Fig 1. Association of CA-AKI and Baseline eGFR.** The participants were stratified according to the grade of AKI depending on their baseline eGFR level: G1 with eGFR ≥ 90 mL/min/1.73 m², G2 between 60 and 89 mL/min/1.73 m², G3 between 30 and 59 mL/min/1.73 m², G4 between 15 and 29 mL/min/1.73 m², and G5 <15 mL/min/1.73 m². Black bars represent the percentage of the participants who did not show AKI and grey bars represent the percentage of the participants who showed AKI.

associated with CA-AKI (aOR = 1.34, 95%CI: 0.96-1.89, and aOR = 2.03, 95%CI: 1.40-2.94, respectively). Patients with chronic kidney disease were about 4 times more associated with CA-AKI (aOR = 3.75, 95%CI: 2.18-6.45). Moreover, patients with CA-AKI were more on loop diuretics (aOR = 2.21, 95%CI: 1.49-3.28), beta-lactams (aOR = 4.11, 95%CI: 2.63-6.42), and allopurinol (aOR = 2.74, 95%CI: 1.43-5.25) (Table 3).

**Table 3. Multivariable logistic regression: factors associated with AKI.**

|  | aOR | 95% C.I. | | P value |
|---|---|---|---|---|
| **Age (≥ 60 years)** | 1.55 | 1.09 | 2.2 | 0.014 |
| **SBP (≥ 140)** | 1.34 | 0.96 | 1.89 | 0.088 |
| **RR (≥ 22)** | 2.03 | 1.40 | 2.94 | <0.001 |
| **Chronic Kidney Disease** | 3.75 | 2.18 | 6.45 | <0.001 |
| **Loop Diuretic** | 2.21 | 1.49 | 3.28 | <0.001 |
| **Beta-Lactams** | 4.11 | 2.63 | 6.42 | <0.001 |
| **Allopurinol** | 2.74 | 1.43 | 5.25 | 0.002 |
| **Vancomycin** | 1.46 | 0.95 | 2.25 | 0.084 |

Variables entered in the model: Age (reference < 60 years), qSOFA (reference: 0), Diabetes, Hypertension, Cardiovascular diseases, Chronic Kidney Disease, Liver disease, Loop Diuretic, Aminoglycoside, Beta-Lactam, Allopurinol, Aspirin, Vancomycin, Colistin, Angiogram CT (reference: No), SBP (reference: <140), and RR (reference: < 22).

Omnibus test <0.001, $R^2$=0.188, Hosmer=0.283; Variables entered < 0.05 in bivariate analysis

Abbreviations: SBP: Systolic Blood Pressure, RR: Respiratory Rate.

## Discussion

Our study identified several factors associated with the development of CA-AKI in patients receiving contrast media. It is essential to note that CA-AKI refers to AKI occurring in temporal association with contrast media administration without implying causation, whereas CI-AKI suggests a direct causal relationship [2]. Given the retrospective design of our study and the absence of a control group, we cannot establish a causative link between contrast media and AKI. This aligns with the guidelines from the American College of Radiology and the National Kidney Foundation, which recommend using CA-AKI to avoid overstating the risk of contrast media [17–19].

There are over 35 definitions of acute kidney injury, but in this study, we adopted KDIGO criteria, which is a widely recognized and accepted international definition of AKI for its standardization, sensitivity, comprehensiveness, evidence-based, and applicability [20]. KDIGO criteria define AKI either by an increase in serum creatinine by ≥ 0.3 mg/dl within 48 hours [21]. We aimed to investigate the factors that are associated with contrast-induced nephrotoxicity (CIN) in an emergency setting at a tertiary care center.

Our study showed that age, hypertension, high respiratory rate, chronic kidney disease, use of loop diuretics, beta-lactam, and allopurinol were significantly and positively associated with the development of CIN. The association between age and CIN is reported in the literature. Elderly individuals face an increased risk because of age-related decreases in kidney function and the existence of various comorbid conditions that can worsen kidney damage. Changes in kidney blood flow and hemodynamics associated with aging, like decreased glomerular filtration rate (GFR) and renal blood circulation, render the kidneys more vulnerable to harmful substances [22]. Recently, Lie et al. showed in their cohort study on 643 patients that age above 70 years is associated with CIN [23]. Additional similar observations were reported by Briguori et al. and Donnarumma et al. [24,25].

Moreover, hypertension is also reported to be a risk factor and is associated with CIN [23,26,27]. Although our results indicated that there is an association between hypertension and CIN, it did not reach statistical significance in our multivariate model. Elevated blood pressure may be indicative of undiagnosed hypertension, a known risk factor for AKI during hospital admissions, as highlighted by Jiang et al. in their study on hypertension and AKI risk [28]. Regarding the high respiratory rate, there are limited reports in the literature on its association with CIN. However, Davenport et al. reported in a retrospective study conducted on more than 20,000 patients that sepsis, which often presents with elevated respiratory rates, is linked to CIN [29]. Moreover, McCullough previously reviewed the association between sepsis and CIN [30]. Given that a high respiratory rate (RR > 22) is a component of the qSOFA score and indicates sepsis, our findings are consistent with existing literature [31].

Chronic kidney disease is another well-established risk factor for CIN. Shinoura et al., in a cohort study on 1475 ED patients, reported that elevated serum creatinine level (above 1.5 mg/dl) before CT with contrast is associated with CIN [32]. Additionally, multiple reports have also identified chronic kidney disease as a risk factor for CIN [33–35].

Previous studies have suggested that the use of angiotensin-converting enzyme inhibitors (ACEIs) and loop diuretics increases the risk of CIN. The proposed mechanism is the reduced intravascular volume, leading to decreased renal perfusion and intrarenal vasoconstriction [36]. Our findings showed that patients on loop antidiuretics were twice as likely to develop CIN. However, we found no significant difference in CIN between those on ACEIs and those who were not.

The literature presents conflicting findings regarding the effect of Allopurinol with CIN. For instance, Rey et al. conducted a post-marketing study to assess the association between

Allopurinol and AKI cases. The findings of this study confirm that Allopurinol increases the odds of developing AKI [37], which aligns with our results. Conversely, many reports in the literature support the idea that Allopurinol decreases the risk of CA-AKI. Ma et al. conducted a meta-analysis on 176 randomized clinical trials and showed that pretreatment with Allopurinol significantly reduced the incidence of CA-AKI (relative risk (RR): 0.37, p = 0.01) [38]. This contrasted with our findings, which showed that patients on Allopurinol were two times more likely to develop CA-AKI.

We also found that using Beta-lactams significantly increased the risk of CA-AKI, with patients on Beta-lactams being four times more likely to develop CA-AKI. This is consistent with studies that have identified nephrotoxic drugs as risk factor for CA-AKI [39,40]. Moreover, numerous publications have highlighted the nephrotoxicity of Beta-lactams antibiotics, particularly on proximal tubular cells [41–46].

Additionally, a cross-sectional study in China found that the concomitant use of vancomycin with contrast material was significantly associated with the development of AKI [47]. This aligns with our results, where Vancomycin use increased the risk of CA-AKI by 1.4 times, likely due to its oxidative effects.

It is important to differentiate between CA-AKI and CI-AKI. CA-AKI is a correlative diagnosis indicating that AKI occurred after contrast administration, while CI-AKI implies a causative diagnosis. This distinction is crucial as our study design does not allow for a causal inference.

## Limitations

As for our study's limitations, This study is a single -center cohort study that relied on data from patients' medical charts. Although we only included the patients who presented to one care center, our Emergency Department is one of the busiest in the country, allowing our results to be extended to other care centers. The retrospective nature of the study limited the authors' ability to have more than one creatinine measurement to check for the AKI definition for some of the patients and include them in the analysis of the study.

On the other hand, the selection bias in our study was reduced by the fact that around 17% of the study participants exhibited an abnormal (above and lower the normal range) baseline creatinine level. Among these 306 participants, only 42 fulfilled the KDIGO definition of AKI. Moreover, our study's lack of a control group limited our ability to reach causative findings. On another note, this study's results pinpoint the need to design and conduct clinical trials to confirm the characteristics of the patients who are protected against CA-AKI and could benefit from rapid CI-CT in acute settings.

## Conclusion

In an emergency setting, assessing the factors associated with CA-AKI, such as age, high respiratory rate, chronic kidney disease, use of Beta-lactams, loop diuretics, and/or Allopurinol could help reduce the need to order serum creatinine lab assessment, expedite the imaging workup, and promptly diagnose life-threatening conditions. Furthermore, our study's results highlight the need to develop clinical trials assessing the safety of administering contrast agents to patients without waiting for their creatinine levels.

## Acknowledgments

The results of this study were presented at the XII[th] Mediterranean Emergency Medicine Congress (MEMC) held between 7 and 10 September 2023 in Rhodes, Greece.

## Author contributions

**Conceptualization:** Moustafa Al Hariri, Imad El Majzoub, Tharwat El Zahran.

**Data curation:** Moustafa Al Hariri.

**Formal analysis:** Moustafa Al Hariri, Malak Khalifeh, Hani Tamim.

**Methodology:** Moustafa Al Hariri, Hani Tamim, Imad El Majzoub, Tharwat El Zahran.

**Resources:** Moustafa Al Hariri.

**Supervision:** Tharwat El Zahran.

**Writing – original draft:** Moustafa Al Hariri, Sally Al Hassan, Malak Khalifeh, Hani Tamim, Imad El Majzoub, Tharwat El Zahran.

**Writing – review & editing:** Moustafa Al Hariri, Sally Al Hassan, Malak Khalifeh, Hani Tamim, Imad El Majzoub, Tharwat El Zahran.

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
