## [Decision Letter · Decision Letter 0]

18 Jul 2024

PONE-D-24-16070Factors Associated with Contrast-Induced Acute Kidney Injury in a Tertiary Care Center Emergency Department: an Observational Cohort Study.PLOS ONE

Dear Dr. El Zahran,

Thank you for submitting your manuscript to PLOS ONE. After careful consideration, we feel that it has merit but does not fully meet PLOS ONE’s publication criteria as it currently stands. Therefore, we invite you to submit a revised version of the manuscript that addresses the points raised during the review process. The term contrast-associated AKI (CA-AKI) is more appropriate and should be used throughout the report. The logistic regression tables should be presented in one table. Clearly state in your statistical plan 1) how predictor variables were selected for the logistic regression analysis 2) How were the confounding variables identified?. Although it is important to include medications and clinical parameters as predictor variables, the authors should check for **multicollinearity** and address them as appropriate. For any two variables with r > 0.8, only one should be included in the table.  In addition, address all concerns raised by both reviewers.

We look forward to receiving your revised manuscript.

Kind regards,

Ogochukwu Chinedum Okoye

Academic Editor

PLOS ONE

Journal Requirements:

2. Please note that in order to use the direct billing option the corresponding author must be affiliated with the chosen institute. Please either amend your manuscript to change the affiliation or corresponding author, or email us at plosone@plos.org with a request to remove this option.

3. In the online submission form, you indicated that "The datasets generated during and/or analysed during the current study are available from the corresponding author on reasonable request."

Reviewers' comments:

Reviewer's Responses to Questions

**Comments to the Author**

1. Is the manuscript technically sound, and do the data support the conclusions?

Reviewer #1: Partly

Reviewer #2: Yes

2. Has the statistical analysis been performed appropriately and rigorously? 

Reviewer #1: No

Reviewer #2: Yes

3. Have the authors made all data underlying the findings in their manuscript fully available?

Reviewer #1: No

Reviewer #2: No

4. Is the manuscript presented in an intelligible fashion and written in standard English?

Reviewer #1: Yes

Reviewer #2: Yes

5. Review Comments to the Author

Reviewer #1: Overview: The authors present the results from a single center, retrospective study of 1,832 emergency department patients who underwent CT with contrast between November 2018 and December 2019. They report a prevalence of contrast-induced acute kidney injury (CI-AKI) of 10.9%, and found that BP≥140mmHg, RR≥22, existing history of CKD, and use of several medications (loop diuretics, beta-lactams, allopurinol) were associated with CI-AKI in a logistic regression analysis.

Major Comments Overall:

- One major issue that must be addressed is that the authors do not differentiate between contrast-associated AKI (CA-AKI) and CI-AKI. CA-AKI, implies a correlative diagnosis of AKI, whereas CI-AKI implies a causative diagnosis. In 2020, the American College of Radiology and the National Kidney Foundation released joint guidelines, published simultaneously in Radiology and Kidney Med (Davenport et al., 2020) that distinguishes between the two. This study must be cited in this paper. The authors methodologic approach does not provide a control group for analysis, therefore no causative link can be drawn between the AKI these patients developed and the contrast they received. Thus, it is crucial that “CI-AKI” be replaced by “CA-AKI” throughout the manuscript. Additionally, this difference should be briefly discussed in the introduction and the discussion, as it helps interpret the authors’ findings.

Abstract:

- Replace “CI-AKI” with “CA-AKI” throughout abstract

- In the conclusions section, you mention “age” as a factor to consider, but this was not significantly associated with CA-AKI in your model. Please remove.

Introduction:

- Please provide citations throughout for each sentence that cites data or concepts from other studies

Methods:

- The study aim should be moved to the last sentence of the introduction

- It is unclear what variables you adjusted for in your logistic regression model: please clarify which variables either in the methods, or in table 2 where you report regression outputs

Results:

- For Table 2, is this a multiple logistic regression model or are these bivariate p values? If it was a multiple logistic regression model, it would be helpful to show the bivariate analysis to see how ORs changed after being included in the multivariate model

- For table 2, it would be helpful to know the row percentages: For example: how many patients with qSOFA 0, 1 or 2 had AKI? How many patients on loop diuretics had AKI compared to those not on loop diuretics? These are the proportions that matter so we can understand how prevalence of AKI differs by each characteristic

Discussion:

- It is interesting that BP and RR are associated with CA-AKI; but how do you suggest interpreting this? It is very likely that elevated BP is simply a marker of undiagnosed hypertension, which itself is predictive of AKI during any admission (Jiang 2021, BMC Nephrology). Similarly, elevated RR may be a marker of acuity of illness; this elevated acuity of illness likely predisposes an increased propensity to AKI regardless of contrast administration. For example, conditions such as volume overload and sepsis would increase the RR and both are known to contribute to AKI. Indeed, it was unsurprising that AKI was more common in the qSOFA 1 group. These diagnoses also increase the likelihood that someone might get a contrast study to evaluate for infection or pulmonary embolism, for example. This must be discussed.

- Must add lack of control group to the limitations part of discussion

Reviewer #2: We would like to thanks to the authors for giving me the opportunity to review the manuscript entitled "Factors Associated with Contrast-Induced Acute Kidney Injury in a Tertiary Care Center Emergency Department: an Observational Cohort Study".

The manuscript is very interesting, but needs a few changes to improve it.

Introduction

Why have the authors placed the objective of the study in the method section? We believe that the objective of the study should be placed at the end of the context.

Method

This study was approved by the IRB,

Authors should explain the abbreviation at the first appearance in the text.

Study approval may be presented in a different section as an Ethics Statement.

The authors must correctly state the inclusion criteria: did the authors include all patients? What is the age criterion for inclusion?

The authors report that patients receive a mean contrast dose of 2ml/kg. As this is an average, could the authors add the standard deviation?

The author should specify the type of iodinated contrast medium used: iso or low-osmolarity?

In the methods section

Outcome measures: this title would normally correspond to data collection.

home medication, vitals, ED management (medications and IV administration), home medications,

In this sentence, home medication is repeated

laboratory workup

The authors should specify which laboratory data were collected and reported in Table 1, where the general characteristics of the study are presented according to AKI status.

Could the authors provide information on the stage of AKI and incidence of hemodialysis in these AKI-IC cases?

Definition of AKI

as an increase in creatinine of 0.3 mg/dl (26 mmol/l) within 48 hours: please correct the unit to 26 µmol/L

The title definition of AKI should be replaced by an operational definition in which the authors could report the definition of Aki and add the definition of CKD.

Statistical analysis

Continuous variables were presented using mean and standard deviation (mean ± SD).

The authors can confirm that all data were normally distributed, as they did not present the results as median (IQR).

Results

In Table 1, what is the difference between antiviral and antiretroviral drugs?

In Table 1, the authors should compare baseline creatinine values between the AKI and non-AKI groups.

Patients who received lithium (30.5% vs. 22%, p=0.008): these results do not correspond to those reported in Table 1.

Patients with high blood pressure (≥140 mmHg): this result is not consistent with that reported in Table 3.

Why were many factors significant in the univariate logistic regression not presented in Table 3 in the multivariate logistic regression?

Please write ˂ 0.001 for the p-value of 0.000.

Limitation

Please correct the KDIGO definition of acute renal failure. For KDIGO

Translated with www.DeepL.com/Translator (free version)

6. PLOS authors have the option to publish the peer review history of their article (what does this mean? ). If published, this will include your full peer review and any attached files.

**Do you want your identity to be public for this peer review?** For information about this choice, including consent withdrawal, please see our Privacy Policy .

Reviewer #1: No

Reviewer #2: **Yes: ** yannick nlandu

---

## [Author Response · Author response to Decision Letter 0]

25 Aug 2024

Dr. Ogochukwu Chinedum Okoye

Academic Editor

PLoS One August 25, 2024

Dear Dr. Ogochukwu Chinedum Okoye,

On behalf of the authors of our submitted manuscript “Contrast-Associated Acute Kidney Injury: an Observational, Cohort Study in an Emergency Department of a Tertiary Care Center” with submission ID PONE-D-24-11651, I would like to thank you and the reviewers for the time to review and suggest improvements for our manuscript. I am submitting herein the point-by-point response addressing the comments and suggestions of the editor(s) and the reviewers.

Dear Dr. El Zahran,

Thank you for submitting your manuscript to PLOS ONE. After careful consideration, we feel that it has merit but does not fully meet PLOS ONE’s publication criteria as it currently stands. Therefore, we invite you to submit a revised version of the manuscript that addresses the points raised during the review process.

The term contrast-associated AKI (CA-AKI) is more appropriate and should be used throughout the report. The logistic regression tables should be presented in one table. Clearly state in your statistical plan 1) how predictor variables were selected for the logistic regression analysis 2) How were the confounding variables identified?. Although it is important to include medications and clinical parameters as predictor variables, the authors should check for multicollinearity and address them as appropriate. For any two variables with r > 0.8, only one should be included in the table. In addition, address all concerns raised by both reviewers.

Response: We thank the reviewer for summarizing the reviewers’ points. We have amended the use of contrast induced AKI into Contrast Associated AKI (CA-AKI) on multiple occasions in the manuscript.

Logistic regression analysis is presented in table 3 of the manuscript. We have reported the

As for the variables that were selected to be tested in the logistic models, we have amended our method section to clarify the selection process and we added to the footnote of Table 3 the variables that were accounted for upon generating the logistic regression model.

As for the collinearity of the variables included in the logistic regression model, we have performed Variance Inflation Factor (VIF) values. In the below table, we present the VIF values of the variables, which were all below 5, indicating a low collinearity between the variables.

Variable VIF value

Age (≥ 60 years) 1.086826

SBP (≥ 140) 1.053905

RR (≥ 22) 1.067888

Chronic Kidney Disease 1.020159

Loop Diuretic 1.142967

Beta-Lactams 1.120602

Allopurinol 1.018106

Vancomycin 1.130351

3. In the online submission form, you indicated that "The datasets generated during and/or analysed during the current study are available from the corresponding author on reasonable request."

Response: We thank the editor and the reviewers for this suggestion. We have deposited our dataset in Qatar University data repository, which is accessible through the following link: https://qspace.qu.edu.qa/handle/10576/57874

The data sharing statement in the manuscript was amended accordingly.

Reviewers' comments:

Reviewer's Responses to Questions

Comments to the Author

1. Is the manuscript technically sound, and do the data support the conclusions?

Reviewer #1: Partly

Reviewer #2: Yes

2. Has the statistical analysis been performed appropriately and rigorously?

Reviewer #1: No

Reviewer #2: Yes

3. Have the authors made all data underlying the findings in their manuscript fully available?

Reviewer #1: No

Reviewer #2: No

4. Is the manuscript presented in an intelligible fashion and written in standard English?

Reviewer #1: Yes

Reviewer #2: Yes

5. Review Comments to the Author

Reviewer #1: Overview: The authors present the results from a single center, retrospective study of 1,832 emergency department patients who underwent CT with contrast between November 2018 and December 2019. They report a prevalence of contrast-induced acute kidney injury (CI-AKI) of 10.9%, and found that BP≥140mmHg, RR≥22, existing history of CKD, and use of several medications (loop diuretics, beta-lactams, allopurinol) were associated with CI-AKI in a logistic regression analysis.

Major Comments Overall:

- One major issue that must be addressed is that the authors do not differentiate between contrast-associated AKI (CA-AKI) and CI-AKI. CA-AKI, implies a correlative diagnosis of AKI, whereas CI-AKI implies a causative diagnosis. In 2020, the American College of Radiology and the National Kidney Foundation released joint guidelines, published simultaneously in Radiology and Kidney Med (Davenport et al., 2020) that distinguishes between the two. This study must be cited in this paper. The authors methodologic approach does not provide a control group for analysis, therefore no causative link can be drawn between the AKI these patients developed and the contrast they received. Thus, it is crucial that “CI-AKI” be replaced by “CA-AKI” throughout the manuscript. Additionally, this difference should be briefly discussed in the introduction and the discussion, as it helps interpret the authors’ findings.

Response: We appreciate the reviewer’s insightful comment. We have addressed this issue by making the following changes to the manuscript:

We have replaced "CI-AKI" with "CA-AKI" throughout the manuscript to reflect the appropriate terminology as per the guidelines from the American College of Radiology and the National Kidney Foundation (Davenport et al., 2020).

We have added the following statement to the Background section to differentiate between CA-AKI and CI-AKI:

" In 2020, the American College of Radiology and the National Kidney Foundation released joint guidelines distinguishing between CA-AKI and contrast-induced acute kidney injury (CI-AKI). CA-AKI refers to AKI that occurs in temporal association with contrast media administration but does not imply causation, whereas CI-AKI implies a direct causal relationship between contrast media and AKI (Davenport et al., 2020)."

We have also added the following sentences at the end of the Introduction section to briefly discuss the difference between CA-AKI and CI-AKI:

"It is important to differentiate between CA-AKI and CI-AKI. CA-AKI is a correlative diagnosis indicating that AKI occurred after contrast administration, while CI-AKI implies a causative diagnosis. This distinction is crucial as our study design does not allow for a causal inference. Therefore, we will use the term CA-AKI in this manuscript (Davenport et al., 2020)."

We have added the following paragraph to the Discussion section to further elaborate on the difference between CA-AKI and CI-AKI and its relevance to our findings:

"Our study identified several factors associated with the development of CA-AKI in patients receiving contrast media. It is essential to note that CA-AKI refers to AKI occurring in temporal association with contrast media administration without implying causation, whereas CI-AKI suggests a direct causal relationship. Given the retrospective design of our study and the absence of a control group, we cannot establish a causative link between contrast media and AKI. This aligns with the guidelines from the American College of Radiology and the National Kidney Foundation, which recommend using the term CA-AKI to avoid overstating the risk of contrast media (Davenport et al., 2023, Abbasi et al., 2022, & Weinreb et al. 2021)."

References:

• Davenport, M. S., et al. (2020). Use of Intravenous Iodinated Contrast Media in Patients with Kidney Disease: Consensus Statements from the American College of Radiology and the National Kidney Foundation. Radiology and Kidney Medicine.

• Davenport MS, Perazella MA, Nallamothu BK. Contrast-Induced Acute Kidney Injury and Cardiovascular Imaging: Danger or Distraction? Circulation. 2023;147(11):847-9. doi: doi:10.1161/CIRCULATIONAHA.122.062783.

• Abbasi N, Glazer DI, Saini S, Sharma A, Khorasani R. Utility of Patient-Reported Risk Factors for Identifying Advanced Chronic Kidney Disease Before Outpatient CT: Comparison With Recent ACR/NKF Consensus Criteria. American Journal of Roentgenology. 2022;219(3):462-70. doi: 10.2214/ajr.22.27494. PubMed PMID: 35383485.

• Weinreb JC, Rodby RA, Yee J, Wang CL, Fine D, McDonald RJ, et al. Use of Intravenous Gadolinium-based Contrast Media in Patients with Kidney Disease: Consensus Statements from the American College of Radiology and the National Kidney Foundation. Radiology. 2021;298(1):28-35. doi: 10.1148/radiol.2020202903. PubMed PMID: 33170103.

Abstract:

- Replace “CI-AKI” with “CA-AKI” throughout abstract

Response: We thank the reviewer for this suggestion. We replaced CI-AKI throughout the manuscript with CA-AKI.

- In the conclusions section, you mention “age” as a factor to consider, but this was not significantly associated with CA-AKI in your model. Please remove.

Response: We thank the reviewer for this suggestion. However, we respectfully disagree. Our logistic regression model, presented in Table 3, indicates that age (specifically, being 60 years or older) is significantly associated with a higher likelihood of developing CA-AKI. The adjusted odds ratio (aOR) for age being 60 years or older is 1.547, with a 95% confidence interval of 1.091 to 2.195 and a p-value of 0.014. Therefore, age should be included in the conclusions as a factor associated with CA-AKI.

Introduction:

- Please provide citations throughout for each sentence that cites data or concepts from other studies

Response: We thank the reviewer for this suggestion. We added citation indications for the data and concepts included in the background section. The added citations are:

3. Andreucci M. Iodinated Contrast-Induced Acute Kidney Injury. Pathophysiology and Prevention. Giornale di Tecniche Nefrologiche e Dialitiche. 2017;29(1):11-9. doi: 10.5301/gtnd.2017.16584.

4. Ye J, Liu C, Deng Z, Zhu Y, Zhang S. Risk factors associated with contrast-associated acute kidney injury in ST-segment elevation myocardial infarction patients: a systematic review and meta-analysis. BMJ Open. 2023;13(6):e070561. doi: 10.1136/bmjopen-2022-070561.

5. Pistolesi V, Regolisti G, Morabito S, Gandolfini I, Corrado S, Piotti G, et al. Contrast medium induced acute kidney injury: a narrative review. Journal of Nephrology. 2018;31(6):797-812. doi: 10.1007/s40620-018-0498-y.

7. Maioli M, Toso A, Leoncini M, Musilli N, Grippo G, Ronco C, et al. Bioimpedance-Guided Hydration for the Prevention of Contrast-Induced Kidney Injury: The HYDRA Study. Journal of the American College of Cardiology. 2018;71(25):2880-9. doi: https://doi.org/10.1016/j.jacc.2018.04.022.

Methods:

- The study aim should be moved to the last sentence of the introduction

Response: We thank the reviewer for the suggestion. We amended the manuscript write-up according to the suggestion.

- It is unclear what variables you adjusted for in your logistic regression model: please clarify which variables either in the methods, or in table 2 where you report regression outputs

Response: we thank the reviewer for this question. We have clarified the variables adjusted for in our logistic regression model as follows:

We have presented the bivariate analysis (chi-square or independent t-test) of our variables in Tables 1 and 2. We also added the False Discovery Rate (FDR) value of the results to account for multiple comparisons.

We assessed and presented the unadjusted odds ratio (OR) of each variable to assist in selecting the variables for the logistic regression model. We relied on the unadjusted OR confidence interval rather than just the p-value or q-value from the bivariate analysis.

The results of the logistic regression are presented in Table 3. The variables selected for the logistic regression model are listed as a footnote underneath Table 3.

The Methods section describing the selection of variables for the logistic regression reads as follows:

“Logistic regression was then done to adjust confounding variables and test the association between CA-AKI and other variables. Variables that showed a significant confidence interval of the unadjusted OR and clinically relevant were selected to build the logistic regression model.”

Results:

- For Table 2, is this a multiple logistic regression model or are these bivariate p values? If it was a multiple logistic regression model, it would be helpful to show the bivariate analysis to see how ORs changed after being included in the multivariate model

Response: We thank the reviewer for this important question. Table 2 presents the bivariate p-values from our analysis and not multiple logistic regression models. We added the unadjusted odds ratio (OR) for each of the variables and their 95% confidence intervals (95% CI) to this table to aid in selecting factors for the logistic regression model presented in Table 3.

To enhance clarity, we have updated Tables 1 and 2 to clearly indicate that they present bivariate p-values and unadjusted ORs. Additionally, we have included footnotes in Tables 1 and 2 to clarify that the ORs and the 95% CIs presented in these tables are generated from unadjusted logistic regression.

The modified footnotes for tables 1 and 2 read as follows:

“Data are presented as numbers with percentages.

The p-value for the difference between AKI vs. no AKI is calculated by chi-square, Fisher´s exact test, or T-test where appropriate.

The q-value is the False Discovery Rate (FDR) significance accounting for multiple comparisons.

The unadjusted Odd Ratio (OR) for each variable was calculated without accounting for any other factor. 95% CI is the confidence interval of the OR for each variable.”

- For table 2, it would be helpful to know the row percentages: For example: how many patients with qSOFA 0, 1 or 2 had AKI? How many patients on loop diuretics had AKI compared to those not on loop diuretics? These are the proportions that matter so we can understand how prevalence of AKI differs by each chara

---

## [Decision Letter · Decision Letter 1]

21 Nov 2024

PONE-D-24-16070R1Factors Associated with Contrast-Associated Acute Kidney Injury in a Tertiary Care Center Emergency Department: an Observational Cohort Study.PLOS ONE

Dear Dr. El Zahran,

Thank you for submitting your manuscript to PLOS ONE. After careful consideration, we feel that it has merit but does not fully meet PLOS ONE’s publication criteria as it currently stands. Therefore, we invite you to submit a revised version of the manuscript that addresses the points raised by reviewer 2. 

Please submit your revised manuscript within Jan 05 2025 11:59PM. If you will need more time than this to complete your revisions, please reply to this message or contact the journal office at plosone@plos.org . Please include the following items when submitting your revised manuscript:

We look forward to receiving your revised manuscript.

Kind regards,

Ogochukwu Chinedum Okoye

Academic Editor

PLOS ONE

Journal Requirements:

Reviewers' comments:

Reviewer's Responses to Questions

**Comments to the Author**

1. If the authors have adequately addressed your comments raised in a previous round of review and you feel that this manuscript is now acceptable for publication, you may indicate that here to bypass the “Comments to the Author” section, enter your conflict of interest statement in the “Confidential to Editor” section, and submit your "Accept" recommendation.

Reviewer #2: (No Response)

Reviewer #3: (No Response)

2. Is the manuscript technically sound, and do the data support the conclusions?

Reviewer #2: Yes

Reviewer #3: Yes

3. Has the statistical analysis been performed appropriately and rigorously? 

Reviewer #2: Yes

Reviewer #3: Yes

4. Have the authors made all data underlying the findings in their manuscript fully available?

Reviewer #2: Yes

Reviewer #3: Yes

5. Is the manuscript presented in an intelligible fashion and written in standard English?

Reviewer #2: Yes

Reviewer #3: Yes

6. Review Comments to the Author

Reviewer #2: We thank the editor for the opportunity to review the revised manuscript.

We feel that the manuscript is improved, but requires some minor change

Introduction

This study aims to assess the factors associated with the development of CA-AKI in patients who received computed tomography (CT) scans with contrast during their presentation to an academic Emergency Department in a tertiary care center. It is essential to differentiate between CA-AKI and CI-AKI. CA-AKI is a correlative diagnosis indicating that AKI occurred after contrast administration, while CI-AKI implies a causative diagnosis. This distinction is crucial as our study design does not allow for a causal inference. Therefore, we will use the term CAAKI in this manuscript [2].

We believe that the second sentence of this paragraph is repetitive and should be deleted. The authors have already correctly distinguished between CA-AKI and CI-AKI in the second paragraph of their introduction.

Methods

Study participants received an average of 2ml1ml/Kg for routine CT and 1.5ml/Kg for angio-CT of low-osmolar contrast medium (Omnipaque 350) contrast solution

In front of 1ml/kg please delete '2ml

Patients’ details were prepared as reports by the EPIC team

Please define the term “EPIC”

Results

Discussion

We also think it would be interesting for the authors of this interesting article to comment on the incidence of CA-AKI reported in their study. In the context where the authors report an association between high respiratory rate and CA-AKI, the higher incidence of CA-AKI in COVID-19 patients (26.1%) (mainly respiratory pathology) could be a further argument in favour of the association between high respiratory rate and CA-AKI. Nlandu Y, Makulo JR, Essig M, Sumaili E, Lumaka A, Engole Y, Mboliasa MF, Mokoli V, Tshiswaka T, Nkodila A, Bukabau J, Longo A, Kajingulu F, Zinga C, Nseka N. Factors associated with acute kidney injury (AKI) and mortality in COVID-19 patients in a Sub-Saharan African intensive care unit: a single-center prospective study. Ren Fail. 2023;45(2):2263583. doi: 10.1080/0886022X.2023.2263583. Epub 2023 Oct 23. PMID: 37870858; PMCID: PMC11001370.

Limitations

On the other hand, the selection bias in our study was reduced by the fact that around 17% of the study participants

exhibited an abnormal (above and lower the normal range) baseline creatinine level. Among these 306 participants,

only 42 fulfilled the KODIGO definition of AKI.

Please correct KODIGO to KDIGO

Reviewer #3: The Abstract and introduction were well written and captivating with minor errors as highlighted in the attachment .

The aim was clear and succinct

Overall methodology was thorough and aligned well with the study aim. However, it may be clearer to structure the paragraph with bullet points or distinct sections for "Study Design", "Setting", Ethical Approval", for better organization.

The results with tables and illustrations was well captured and discussion was sound as well.

By and large, very fine manuscript.

7. PLOS authors have the option to publish the peer review history of their article (what does this mean? ). If published, this will include your full peer review and any attached files.

**Do you want your identity to be public for this peer review?** For information about this choice, including consent withdrawal, please see our Privacy Policy .

Reviewer #2: No

Reviewer #3: No

---

## [Author Response · Author response to Decision Letter 1]

12 Dec 2024

We thank the editor and the reviewers for sharing their suggestions to improve our manuscript “Factors Associated with Contrast-Associated Acute Kidney Injury in a Tertiary Care Center Emergency Department: an Observational Cohort Study” with submission number “PONE-D-24-16070R1.”

We would like to indicate that the title of the manuscript was modified according to the suggestion on the provided manuscript word document file to the following:

“Factors Associated with Contrast-Associated Acute Kidney Injury in an Emergency Department: A Cohort Study in Lebanon.”

We have reviewed all the comments added to the document and implemented the needed changes. All changes were tracked on the file. Additionally, please find below our point-by-point response to the points raised by the reviewers.

Reviewers' comments the Author:

Reviewer #2: We thank the editor for the opportunity to review the revised manuscript.

We feel that the manuscript is improved, but requires some minor change

Introduction

This study aims to assess the factors associated with the development of CA-AKI in patients who received computed tomography (CT) scans with contrast during their presentation to an academic Emergency Department in a tertiary care center. It is essential to differentiate between CA-AKI and CI-AKI. CA-AKI is a correlative diagnosis indicating that AKI occurred after contrast administration, while CI-AKI implies a causative diagnosis. This distinction is crucial as our study design does not allow for a causal inference. Therefore, we will use the term CAAKI in this manuscript [2].

We believe that the second sentence of this paragraph is repetitive and should be deleted. The authors have already correctly distinguished between CA-AKI and CI-AKI in the second paragraph of their introduction.

Response: We thank the reviewer for the suggestion. We removed the repetitive statement.

Methods

Study participants received an average of 2ml1ml/Kg for routine CT and 1.5ml/Kg for angio-CT of low-osmolar contrast medium (Omnipaque 350) contrast solution

In front of 1ml/kg please delete '2ml

Response: We thank the reviewer for the suggestion. The edits are implemented in the method section.

Patients’ details were prepared as reports by the EPIC team

Please define the term “EPIC”

Response: We thank the reviewer for the suggestion. EPIC is the name of the Electronic Health record used in our institution. Therefore, it is defined as such is the method section “Electronic Health Record (EPIC) team .”

Results

Discussion

We also think it would be interesting for the authors of this interesting article to comment on the incidence of CA-AKI reported in their study. In the context where the authors report an association between high respiratory rate and CA-AKI, the higher incidence of CA-AKI in COVID-19 patients (26.1%) (mainly respiratory pathology) could be a further argument in favour of the association between high respiratory rate and CA-AKI. Nlandu Y, Makulo JR, Essig M, Sumaili E, Lumaka A, Engole Y, Mboliasa MF, Mokoli V, Tshiswaka T, Nkodila A, Bukabau J, Longo A, Kajingulu F, Zinga C, Nseka N. Factors associated with acute kidney injury (AKI) and mortality in COVID-19 patients in a Sub-Saharan African intensive care unit: a single-center prospective study. Ren Fail. 2023;45(2):2263583. doi: 10.1080/0886022X.2023.2263583. Epub 2023 Oct 23. PMID: 37870858; PMCID: PMC11001370.

Response: We thank the reviewer for the suggestion. We would like to emphasize that the study was conducted before the COVID period, and no COVID-19 patients were included in the study. Therefore, we will not compare our results to the ones reported in the proposed study.

Limitations

On the other hand, the selection bias in our study was reduced by the fact that around 17% of the study participants exhibited an abnormal (above and lower the normal range) baseline creatinine level. Among these 306 participants, only 42 fulfilled the KODIGO definition of AKI.

Please correct KODIGO to KDIGO

Response: We thank the reviewer for the suggestion. We corrected this typing mistake.

Reviewer #3: The Abstract and introduction were well written and captivating with minor errors as highlighted in the attachment .

The aim was clear and succinct

Overall methodology was thorough and aligned well with the study aim. However, it may be clearer to structure the paragraph with bullet points or distinct sections for "Study Design", "Setting", Ethical Approval", for better organization.

The results with tables and illustrations was well captured and discussion was sound as well.

By and large, very fine manuscript.

Response: We thank the reviewer for the revision.

---

## [Editor Report · Decision Letter 2]

15 Dec 2024

Factors Associated with Contrast-Associated Acute Kidney Injury in an Emergency Department: A Cohort Study in Lebanon

PONE-D-24-16070R2

Dear Dr. El Zahran,

We’re pleased to inform you that your manuscript has been judged scientifically suitable for publication and will be formally accepted for publication once it meets all outstanding technical requirements.

Kind regards,

Ogochukwu Chinedum Okoye

Academic Editor

PLOS ONE
---

## [Editor Report · Acceptance letter]

PONE-D-24-16070R2

PLOS ONE

Dear Dr. El Zahran,

I'm pleased to inform you that your manuscript has been deemed suitable for publication in PLOS ONE. Congratulations! Your manuscript is now being handed over to our production team.

Kind regards,

on behalf of

Dr. Ogochukwu Chinedum Okoye

Academic Editor

PLOS ONE